# Designing Away Waste: A Comparative Analysis of Urban Reuse and Remanufacture Initiatives

**Isabel Ordóñez** [1],*, **Oskar Rexfelt** [2], **Shea Hagy** [3] **and Luisa Unkrig** [1]

1    Department of Environmental Technology, Technische Universität Berlin, D-10623 Berlin, Germany;
     luisa.unkrig@yahoo.de
2    Department of Industrial and Materials Science, Chalmers University of Technology,
     412 96 Gothenburg, Sweden; rex@chalmers.se
3    Department of Architecture and Civil Engineering, Chalmers University of Technology, 412 96 Gothenburg,
     Sweden; shea.hagy@chalmers.se
*    Correspondence: isabel.ordonez@tu-berlin.de

**Abstract:** In order to transform the economy into one that is circular, that recovers most materials through reuse, remanufacturing and recycling, these activities need to grow significantly. Waste management has substantially incorporated recycling as an end-of-life treatment but has still largely failed to incorporate remanufacturing and reuse as possible material recovery routes. This article aims to provide useful information to establish centers for urban remanufacture (CUREs), by analyzing fifteen existing initiatives that facilitate reuse and remanufacture by providing access to secondary materials or manufacturing tools. The study consists of a review of selected initiatives complemented with targeted interviews to fill in missing information. Most initiatives provided access to secondary materials (13 of 15 initiatives), and almost all used different manufacturing tools (14 of 15 initiatives). Besides their regular opening hours, initiatives were mainly engaged in capacity building activities, which were done through predefined or improvised workshops. Most initiatives relied on external support to finance their operations (9 of 15 initiatives). However, one of the self-financed initiatives is the oldest initiative in the study, operating since 1998. Based on the results and tacit knowledge collected in this study, a framework is suggested to serve as a guide for establishing future CUREs.

**Keywords:** reuse; remanufacture; circular economy; maker movement; up-cycling; grass-root initiatives

## 1. Introduction

In order to achieve a resource efficient economy, production and waste management systems need to change so as to significantly increase material recovery. Sustainable waste management has strived to incorporate recycling as a desirable end-of-life treatment, but has so far largely failed to incorporate remanufacturing and reuse as possible material recovery routes. For sustainable waste management to include these actions as desirable recovery routes at the end-of-life stage (or "end-of-current-life" stage), a much better understanding is needed of how these activities are done in practice, how they can be standardized and how they affect the environment. The aim of the study presented in this paper is to report how a number of urban reuse and remanufacturing initiatives operate, how they are financed, to what extent they contribute to resource recovery and what common lessons and themes can be identified between them.

### 1.1. A Global-Local Challenge

One of the main challenges in connecting production and waste management systems is that they operate at different scales, making it a "global-local" problem [1]. On one hand, the production system

is global, with manufacturing mostly done in Asia, Europe and North America, using virgin raw materials from around the globe to supply finished goods for the global market. On the other hand, waste management operates locally, with local authorities charged with coordinating the collection and treatment of residues, ideally following national and regional legislation.

International trade currently spans the globe, with product value chains extending over several continents. Generally speaking, each stage of the value chain adds economic value to the item being developed, investing labor and energy, while generating some type of production waste. Commonly, the energy needed and environmental impacts generated for raw material extraction are much larger than for the later stages of manufacture and sales for consumption, while the value added is higher at the later stages of the chain [2]. This generates an international trade system where raw material exporters are left with less added value, and most of the direct negative environmental effects. In contrast, manufacturing nations are characterized by generating high added value, employment and other positive side effects. The world's regions still have a large divide between raw material providers and manufacturers, where between 75–85% of regional exports for North America, Europe and Asia are in manufactured goods, while 70–80% of the exports from Latin America, Middle East and Africa are minerals, fuels and agricultural products [3]. Economic globalization has made it possible to relocate several key manufacturing sectors from developed to developing regions, giving China the status of the "world's manufacturing center", with the economic growth and environmental problems that implies [3].

Municipal solid waste (MSW) generation correlates strongly with national income: the more a country earns the more solid waste it generates. However, MSW generation rates in high-income countries have started to level out or even show a small decrease that might hint to a decoupling of waste generation from income. This is only observed in high-income nations, and given that economies will continue to grow in medium and low-income countries, the global per capita waste generation is still expected to significantly rise [4]. Additionally, it is more common that high-income countries have extensive waste collection and treatment services, with a steady decline of landfilling, that favor more material recycling and energy recovery. Originally, recycling programs in Europe and North America used recycling markets and industries relatively close to their source. However, the trade of secondary materials has emerged as a global business in the last twenty years, making secondary material prices volatile on the global market, which in turn impedes planning for recycling [5]. Regardless of how global recycling has become, waste collection and initial sorting is by definition a highly localized process. How the generated waste flows are to be sorted and collected is subjected to local regulations, which in some cases follow larger regional mandates (e.g., European Waste Directive and how it is in turn implemented by each member state). However, just the harmonization of waste management and extended producer responsibility (EPR) data has proven to be a long, still on-going process within the European Union (EU) [6].

*1.2. Resource Recovery Routes*

Reuse is a wide spread activity, done both at an individual level (i.e., when users pass on their used belongings to others that will continue to use them), as well as through intermediating institutions, (e.g., second-hand shops, used good web portals and dealers). It is widely promoted as a waste prevention strategy [4,7], however it is not always evident that the reuse of an item will result in environmental benefits, depending on whether or not the reused item replaces the sale of a new product, if it is at least as energy efficient as its new counterparts, and whether the item would have been landfilled, incinerated or recycled [8]. However, reuse is generally a more favorable option than recycling, since it preserves a lot of the energy that was originally used to manufacture the item. As an example, Gothenburg in Sweden currently operates five recycling centers of which one also focuses on reuse. If instead all five centers would do this, 2200 tons of waste could be prevented each year, equivalent of 8000 tons $CO_2$ [9].

Some definitions of reuse include the use of recovered components or materials in the production of new or repaired products. This is in some cases also referred to as remanufacturing. This article uses the definitions that associate material recovery routes to the life cycle stages of a product, i.e., reuse takes materials back to the use phase, remanufacture brings materials to the manufacturing stage (i.e., industrial or semi-industrial process that makes a finished good.), and recycling recovers resources through new material production (as defined in [10]). Remanufacturing tends to be more labor intensive than manufacturing with virgin materials, because the recovered materials or components need to be prepared for reuse and checked if they are suitable for the intended use. Because of this, it makes more commercial sense to remanufacture large, highly valuable goods [8]. Furthermore, remanufacturing has a higher barrier to entry than recycling [11], since the flow of products needs to be heavily controlled. Another often mentioned challenge associated with remanufacturing is the risk of cannibalizing on new product sales (e.g., [12]), even if other researchers argue that the risk may not be as big after all [13]. The most commonly mentioned success cases for remanufacturing refer to high value products in a business to business model, e.g., Ricoh printers and Caterpillar machinery [14,15]. In these cases, the original manufacturing industries take back their own products and do product development planning to facilitate this process. This can be denominated as 'planned remanufacturing', and it is what is often referred to only as 'remanufacturing' and not differentiated from other remanufacturing processes in the literature. Despite the interest in remanufacturing as a key circular economy practice, it still accounts only for ~2% of US production and ~1.9% of EU production [16].

However, recovered materials and components can be also used by independent manufacturers [17] to make similar or different types of products. This can be denominated as 'improvised remanufacturing' given that the remanufacture step is not planned for during the original product development. In the literature this approach is commonly referred to as component, product or material reuse, repair, refurbishment or up-cycling. This approach has a long history with numerous examples of up-cycled products both by amateurs and professionals, with results varying in terms of quality and scalability (e.g., [18–21]).

### 1.3. From Consumers to Pro-Sumers and Local Production Systems

In recent years, the proliferation of repair cafes, hackerspaces and open workshop spaces such as bike kitchens, reflect a growing interest in retaining products in use longer, if they are possible to repair and refurbish. These spaces have appeared mainly organized by volunteers, where the users come to lend appropriate tools and learn how to fix a specific item [22]. In addition, the maker movement (i.e., the culture associated to the rise of hackerspaces, Fab Labs and other "makerspaces" [23] has under recent years spread out globally and enabled local makers to engage in rapid digital prototyping and small-scale manufacturing. Some authors have suggested that the availability of low-cost digital production facilities will significantly change the way we make and consume goods, regarding it as the new industrial revolution [24]. However, these spaces are still used by a niche audience interested in tinkering, and massive changes in production and consumption still remain to be seen.

The European Union's decision to strive for the implementation of a circular economic model requires systematic efforts to better understand and promote reuse and remanufacturing activities. The European Commission has already declared targets to improve the uptake of secondary materials in the EU, develop secondary material standards, while still controlling the presence of substances of concern in the recovered materials [25]. In addition, article 11 of the Waste Framework Directive states that "member States shall take measures, as appropriate, to promote the re-use of products and preparing for re-use activities, notably by encouraging the establishment and support of re-use and repair networks . . . " [26]. The idea of establishing experimental material recirculation hubs in urban areas has been proposed as a way to facilitate the transition to a sustainable material management model, that would help increase knowledge about the secondary material currently available [1,27]. Such hubs are needed to provide space for experimenting on selected recovery pathways, bringing

actors from different sectors together in a space for collaboration, where changes in practice and perspective can be attained. There is a need to have a place to experiment with circular production and consumption solutions that can then be developed and matured locally to provide an alternative circular economic system. These spaces would benefit from combining the creative open innovation culture typical of the maker movement, with access to locally available secondary materials.

*1.4. Aim and Limitations of This Study*

In order to inform how experimental material recirculation hubs could operate, this article analyzes existing initiatives that engage or promote reuse and/or remanufacture by providing access to secondary materials and/or manufacturing tools. This study and the identified existing initiatives were limited to three main urban centers in Northern Europe; Berlin, Germany, Gothenburg Sweden and Copenhagen, Denmark. This geographical limitation was intentional in order to ensure that the initiatives could either be visited in person by the authors and/or a personal interview could be conducted. This study is exploratory in nature and focuses primarily on practitioners' knowledge, as there were few comparable studies found in the literature. Therefore, this study is limited to local knowledge and networks in the three respective geographic locations, and does not include an analysis of all existing initiatives but presents a number of relevant examples in each location. This study aims to identify potential underlying themes among the selected initiatives that may facilitate the implementation of future recirculation hubs, by collecting some of the tacit knowledge such initiatives have. However, it is important to note that the implementation of such hubs is highly dependent on local laws, and regulations, as well as cultural, social and economic aspects.

## 2. Materials and Methods

The present study is part of a larger project that wishes to establish centers for urban remanufacture (CURE). These centers would constitute both a physical place and a network of actors that provide experiments, and eventually reuses or remanufactures secondary materials [28]. The present study is a part of the CURE Pathfinder project financed by EIT Climate-KIC, between August and December, 2018. It was intended to provide an overview of existing initiatives that could provide lessons relevant to implementing CUREs in different cities, with a main focus on Berlin, Germany, Copenhagen, Denmark and Gothenburg, Sweden, where the Pathfinder aimed to contact actors to establish pilot CUREs. In the study, data was collected to inform how such initiatives operate, how they are financed, to what extent they contribute to resource recovery and what common lessons and themes can be identified. Given that the study was done under limited time, it was decided to focus in-depth on 15 relevant initiatives, in or close to the targeted cities. Therefore, to provide an overview of all relevant initiatives in these locations was beyond the scope of this study.

The initiatives (see "Results" for a more detailed descriptions of them) were selected if they made use of or provided secondary materials and/or manufacturing tools. The initiative selection process was as follows: First, one initiative that both used secondary material and had manufacturing tools was identified in Copenhagen (i.e., Guldminen). Then, three initiatives with a physical warehouse that sold or provided secondary material were selected, in Gothenburg (i.e., Återbruket), Copenhagen (i.e., Genbyg), and Berlin (i.e., Material Mafia). Later, local maker spaces and Bike Kitchens were identified in Gothenburg and Berlin (i.e., Fab Lab Berlin, Happylab Berlin, Mikrofabriken, Bike Kitchen North East, and Cykelköket). From there on, a snowballing technique was used, asking the initiatives already selected of other projects that they could recommend for us to include in our study. The suggested initiatives that provided different inputs to the study than the ones already identified, were included in the final analysis.

The data collection began with collecting all information available on the initiatives through their official web-pages, which was later complemented and expanded by targeted semi-structured interviews, to provide answers on topics that were not covered by the available information. A questionnaire guide was created and used to collect the relevant information for each initiative.

The questions were used as a semi-structured interview guide when contacting the initiatives directly. The study realized a total of twelve interviews, one per initiative, with three exceptions: 1) Genbyg, was contacted too late in the process so an interview was not possible in the study time frame, 2) Stilbruch and 3) the BSR exchange platform, both of which had enough information on-line to respond all the questions, and given the time constraints were not contacted. Of the twelve interviews made, ten were done in person and two over the phone. They ranged between 30 and 90 min; notes were taken on the spot and, when allowed by the interviewee, they were also recorded. Answers were collected in full detail into a review chart, that was later categorized into recurring themes for each question. These themes provide the basic structure for this article.

## 3. Results

The initiatives analyzed are described in Table 1. Respecting the differences of how these initiatives operate, the results are presented under four main sections, two of which are derived from the criteria of selecting the initiatives (i.e., access to secondary materials and/or workshop spaces) and the remaining two are used to describe how the initiatives operate (i.e., what activities they do and how they are financed). These four sections are described in more detail in the following text, where they highlight what underlying themes have been identified for each section based on the study results.

**Table 1.** Overview of initiatives reviewed.

| Name | Location & Space | Type of Organization | Main Activities and Results | Running Since |
|---|---|---|---|---|
| 1. Guldminen | ~400 m$^2$ warehouse and workshops space in the recycling center Vasbygade, Copenhaguen | 3-year project run by the Municipality, to test if they can reuse material from recycling centers | Municipality chose 12 "Gold-digger" projects that worked with reuse and recycling. Each Gold-digger group was a self-organized initiative, and in total, during the first year they used 1% of the material received at the recycling station | 2015–2018 |
| 2. Återbruket | ~1800 m$^2$ warehouse located at the recycling center in Alelyckan, Gothenburg | Not for profit, secondary material warehouse run by the Municipality. 5 employees, plus trainees and support staff that add to an additional 3.5 full time staff | Sort, offer and sell incoming material left at the recycling center. The workers ask people who bring material if it could be reused, if this is allowed, then the material gets sorted in to a normal second-hand shop located on-site, or to Återbruket, that focuses specifically on selling used construction material and other building accessories | 2007 |
| 3. Genbyg | ~1000 m$^2$ warehouse in Kastrup, Copenhagen | Business that sells secondary material, mostly construction material | Source, sort, offer and sell secondary material. They also have an updated on-line catalog. In 2012 they also started designing their own products with their material | 1998 |
| 4. Material Mafia | ~150 m$^2$ warehouse in Berlin | Non for-profit business, that promotes zero waste and sells secondary materials | Source, sort, offer and sell secondary material. Offers also workshops and events around waste prevention, reuse and remanufacture. Does product development on demand. | 2012 |
| 5. Fab Lab Berlin | ~500 m$^2$ in Prenzlauer Berg, Berlin | Social community, with membership fees to use the available machines, subsidized mainly by the company Ottobock | A space for people to work with latest production technologies, learn new tools, meet like-minded people and develop their own projects. | 2013–2018 |
| 6. Happylab Berlin | ~300 m$^2$ in Mitte, Berlin | Social community, with membership fees to use the available machines | Provide modern digital fabrication tools and spread their use. To make machines and replicate the knowledge.[1] | 2016 |

**Table 1.** *Cont.*

| Name | Location & Space | Type of Organization | Main Activities and Results | Running Since |
|---|---|---|---|---|
| **7. Mikro-fabriken** | ~1000 m$^2$ in Hisingen, Gothenburg | Commercial maker space, with membership fees to use the machines | Encourage people to create. Enable members to work on their own projects. Targeted for professional makers. | 2015 |
| **8. Bike Kitchen North East** | ~40 m$^2$ in Weissensee, Berlin | Non for-profit group that aims to teach and help people repair bicycles | Collect donated bikes, organize workshop space, helping people with any bike problems, stripping bikes for useful parts, keeping open hours, participating in events and promoting bike culture in the city. [1] | 2013 |
| **9. Cykelköket** | ~60 m$^2$ in Linne, Gothenburg | Non for-profit association that aims to teach and help people repair bicycles | | 2012 |
| **10. Sekundär-Schick** | Based in Berlin from a private workshop | Self-employed person dedicated to promote remanufacturing with used textiles | Teaches people to reuse clothes by remanufacturing them, giving the means so people can update and change clothes themselves | 2010 |
| **11. BSR exchange and donation Market** | Digital exchange platform for Berlin | Digital platform aimed to reduce bulky waste, subcontracted | A product exchange platform managed by the Berlin Waste Management company BSR | 2004 |
| **12. Re-create Design** | ~200 m$^2$ workshop in Gothenburg | Interior design company. 2.5 employees plus 5 trainees | They extend the lifespan and increase value of existing materials, by doing reuse-focused interior design projects, inventory relocation projects, and various consulting activities | 2011 |
| **13. Fixoteket** | 4 open community spaces, between 100–145 m$^2$ in different districts in Gothenburg | Two-year project run by the Municipality in collaboration with local housing associations, to test if such spaces can facilitate sharing, reuse, repair, and recycling of hazardous waste | Neighbors can exchange and repair items, lend tools and participate in organized events around waste prevention. The project hopes to find a way for the spaces to continue to operate after the project period is done | 2017–2019 |
| **14. Stilbruch** | ~3500 m$^2$ second-hand shop in three districts of Hamburg | Not for profit second-hand shop run by the Municipality of Hamburg to reduce waste. 74 employees | Sort, offer and sell second hand furniture and household items. Main tasks include: organizing workshop space, having open hours, repairing items if needed, checking electrical appliances. | 2014 |
| **15. GreenFabLab** | ~500 m$^2$ labs in Cerdanyola des Vallès, Barcelona | Social community, with membership fees to use the machines, associated to the Institute for advanced architecture of Catalonia | This open fabrication lab focuses on creating a self-sufficient habitat and research center, where it is possible to learn from nature to regenerate 21st-century cities. Work is done on growing and managing renewable materials locally and using secondary materials to develop products. | 2011 |

[1] Both of these initiatives engage in the same type of activities.

Although this selection of initiatives is rather heterogeneous, one that stands out and needs further description is Guldminen in Copenhagen, as it works as an umbrella for twelve different projects. Each of these so-called "Gold-digger" projects reuse and remanufacture the material from the recycling center in different ways. For example, the gold-digger association Materiale Centralen categorizes and stores secondary material, which it then donates for others to remanufacture. Another gold-digger initiative collects smaller materials to do creative workshops with school children. The Gold-digger projects were allowed to take any material left at the recycling station, with the exception of the hazardous waste such as electronics.

After the first year of operation of Guldminen, the different projects were allowed to have their own container on site, where they could ask for specific materials that they would reuse in their projects. For example, one project asked for stuffed animals, that came in large amounts, as a stream of their own. Another project asked for electronic items in working condition. Because these items were donated specifically as working electronics, they were not classified as electronic waste and could later be cleaned, repaired partially, if needed, and then sold.

By the end of the Guldminen project, it was unclear how the project or initiatives would continue to operate. All initiatives were independently organized. Some initiatives generated revenue for themselves, while others did not, operating entirely on a voluntary basis. Besides a loose collaboration between the initiatives and the municipality, there was no reliable organizational structure and no clear leadership suited to continue the project further. Some initiatives will continue on their own, while others might not.

### 3.1. Facilitating the Recovery of Secondary Materials

Thirteen of the fifteen initiatives analyzed actively engage in facilitating the reuse of secondary materials. When analyzing their activities, we can see two main aspects that vary among these initiatives: what materials they focus on, and how they enable recirculation. Table 2 organizes these 13 initiatives in a matrix according to these two criteria. The criteria are then explained further in the text, together with other aspects relevant to the initiatives that recovered secondary materials.

**Table 2.** Initiatives organized after type of material used and how they support reuse.

|  | Direct Reuse | Remanufacture for Others | Own Product Development |
|---|---|---|---|
| **Wide range of materials** | Guldminen (CPH)[1]<br>Återbruk (GBG)<br>Genbyg (CPH)<br>Material Mafia (BER) |  | Re-Create Design (GBG)<br>Green Lab (BCN)<br>Mikrofabriken (GBG)<br>Genbyg Design (CPH)[2] |
| **Miscellaneous household items** | Fixotek (GBG)<br>Stilbruch (BER)<br>BSR Exchange(BER) |  |  |
| **Bikes** | Cykelköket (GBG)<br>Bike Kitchen North East (BER) |  |  |
| **Clothing** |  | SekundärSchick (BER) |  |

[1] Considering some of the gold-digger projects. [2] Design company established by Genbyg in 2012.

### 3.1.1. Differing Materials and Material Sources

The materials that the initiatives in this study focus on varies and includes household items, clothing, bikes, and a broad range of other materials. When analyzing the initiatives, it is difficult to draw a strict line between products and materials, since discarded products are sometimes fixed to working condition and reused, but other times are used as a material to make new products (e.g., trays made from blackboards sold by Genbyg).

Four initiatives provide access to a wide range of materials for direct reuse or remanufacture by other actors. The materials they provide include construction material, wood, metal, glass, ceramics, plastics, textiles and composites in large to medium formats, as well as a variety of products, that are normally not found in second hand shops, e.g., ladders, outdoor furniture, wall mirrors, props from theater or film, and bathtubs among other items. Two of these initiatives (i.e., Guldminen and Återbruk) are located directly at a recycling center and obtain their material directly from what is brought in. The rest of the initiatives rely on direct material donations from their collaborating network, e.g., Material Mafia gets material donations from different film and theater productions as well as local industries, while the bike kitchens get old bikes donated to them by housing companies or neighbors. Like the

materials brought to the recycling centers, donated materials vary widely, not only in the materials and products offered, but also in the quality these items have. Donations are sometimes directly delivered, and sometimes offered to the initiatives to collect. A wide variety of materials are also used by the initiatives that engage in their own product development.

The initiatives that facilitate the reuse of miscellaneous household items also include bikes and clothing as part of their material offer. However, they do not specialize on them. In contrast, both bike kitchens and SekunärSchick focus exclusively on their materials, specializing in their repair and remanufacture, providing the branch specific knowledge and consumable materials that are needed.

### 3.1.2. Differing Recovery Routes

The initiatives that work with recovering secondary material do so, in three different ways. Namely, they:

- Offer materials to other actors for direct reuse (e.g., exchanging used items).
- Offer materials for other actors to remanufacture (e.g., used doors made into tables by others).
- Use secondary material in their own product development through remanufacture (e.g., remanufactured furniture).

These different ways of recovering material depend on the quality and type of items that are going to be recovered. For example, if household items are in good condition, they can easily be relocated to a new user, with no need to do any alterations. This relocation can be done through initiatives, such as Stilbruch, Fixoteket or the BSR exchange platform, or any other second-hand stores for that matter. However, if the item to be recirculated is half a fiber-wood board, or a 2 m long wooden beam, they will most likely be altered by the new user, in some sort of remanufacturing process, to fit their new use/purpose. Likewise, if the item to be recovered is a table with a broken leg, it will have to be altered, either through a straight forward repair of the same table, or using it as part of something else.

The difference between repair and remanufacture is not always clear. In the case of both Bike kitchens (i.e., Cykelköket and Bike Kitchen North East), volunteers help people fix old bikes so they can be used again. Some used bikes may just need their tires to be pumped up and their break systems slightly adjusted, to be ready to be used again. However, some used bikes may need more substantial work. If in the process of fixing a bike, crucial functional elements are changed (e.g., including more gears, changing the frame, or shifting pedal breaks to hand breaks), it could be argued that it is not the same bike that is reused, but rather, it has been remanufactured into a different product.

Initiatives that recirculate materials in large format, are more likely to recover for remanufacturing, rather than direct reuse. However, Återbruket, Genbyg and Material Mafia also offer products, like wardrobes, tables or mirrors, which might be directly reused. These three initiatives have seen how their clients make creative use of the material that they provide, and have also engaged in remanufacturing themselves on occasions. Genbyg is so far the only one of these initiatives that started their own offer of remanufactured products.

The initiatives that use secondary material to do their own product development, work more like design firms or manufacturing spaces, than material marketplaces. In this review, two of these initiatives make it easy for their public to engage in remanufacturing with secondary materials (i.e., Green Lab and Mikrofabriken), while Re-create design and Genbyg Design focus on selling their own creations.

### 3.1.3. Inventory Systems

Only six out of the fifteen initiatives keep an inventory of sorts. The other initiatives are either too small to keep an inventory or did not find any use for it. Even if there is an inventory the way it is approached differs greatly. Each initiative has defined on their own what material or product categories to use, based on the material they get but also on their distinctive form of organization. For example, the BSR exchange platform registers the number of items put up on the web-page for each category.

They use ten categories that vary between furniture, appliances, items for kids, hobby, etc. while Återbruket uses their cash flow to keep some sort of overview of their inventory. However, in both cases, the use of an "other" category may bring in confusion when analyzing the statistics. Guldminen asked its gold-miner projects to estimate the amount of material they recovered by providing a form for them to fill out, but how each initiative accounted for their recovered material is unclear.

### 3.2. Facilities and Tools

With the exception of BSR's digital exchange platform, all initiatives reviewed had some sort of workshop space with tools. The main differences between these workshop spaces is their availability to the public and the type of tools they have. Some workshops were used exclusively by the initiative for their own work, while others were open to public on special occasions, or received general public regularly.

### 3.2.1. Workshop Availability to the Public

Closed workshops (five of the fourteen initiatives) have workshop spaces that are not publicly available. Two of these workshops are used exclusively for minor fixes to the items that come in (i.e., Stilbruch and Återbruket), while the workshops of Genbyg and SekundärSchick were used by the initiatives exclusively to do their own production.

As mentioned earlier in the first part of Section 3, the Guldminen project gave access to the warehouse with workshop spaces only to a few selected gold-digger projects. There were no official open hours where people could visit this space, unless it was organized by one of the gold-digger projects or the Copenhagen municipality.

Four initiatives had partially open workshops. On occasions Material Mafia, Re-Create Design and Mikrofabriken will organize events, where participants are welcomed into their workshop spaces for a specific activity. Painting or building workshops, and corporate team building are some examples of such activities. In these cases, the events require preregistration and have a maximum number of participants. Re-Create Design and Mikrofabriken have a fixed workshop space that they open on these occasions. In the case of Material Mafia, the secondary material warehouse is re-organized to house a space to use the tools that the initiative can provide.

Another example of a partially open workshop is the Green FabLab. One reason is that it is associated to the research Institute for advanced architecture of Catalonia. Because of this research orientation it limits visitors into their facilities only after pre-booked appointments. Also, it is not as easily accessible to the public, given that it is located in a large forest near Barcelona.

Five initiatives had open workshops. Bike kitchens and FabLabs are in principle open workshop spaces dedicated to bikes and digital prototyping respectively. Even though the Green FabLab does not focus on being open to general public all the time, both of the other FabLabs in this review do (i.e., FabLab Berlin and Happy Lab Berlin). Both these initiatives provide open hours with personnel that can guide and explain to visitors how to use the space and the machines. In both cases, you have to pay a fee to use the machines, but anyone walking through the door can become a member on the spot and use to use the workshops.

In the case of the Bike kitchens included in the study (i.e., Cykelköket and Bike Kitchen North East) it is not even necessary to become a member to use the workshop space during open hours. Both initiatives rely on volunteers to manage the open hours, helping visitors find the necessary tools and fix their bikes. If a visitor becomes more engaged and starts to volunteer in the space, then they might also be entitled to use the space on their own, on other time slots.

Likewise, Fixoteket has open hours with personnel. However, only some of the personnel are hired by the Municipality through the project funding, while others are part of volunteer organizations that help keep these community spaces open longer.

3.2.2. Types of Tools and Machinery

All workshops are equipped with basic tools like screwdrivers, saws, and cutting knives. The tools available in the mentioned workshops mostly follow the aim of each initiative. For example, both bike kitchens have tools specifically for fixing bikes, while SekundärSchick has sewing machines and dying equipment.

All the FabLabs included in the study provide a similar range of tools. In addition to the general tools they also have CNC mills, laser cutters, 3D printers, electronics workshops, computers and even some robotic arms. FabLab Berlin also offers sewing and knitting machines, that can work with digital or semi digital input.

Material Mafia, Guldminen and Stilbruch have woodworking tools, since it is a commonly used material within their projects. Re-Create design on the other hand uses mostly paint and brushes in their remanufacturing.

*3.3. Events and Activities*

All initiatives engaged in some type of events with participants, with the exception of BSRs digital exchange platform. Nine of the reviewed initiatives offer some sort of workshop or teaching activities (i.e., Fixoteket, Fab Lab Berlin, Happy Lab Berlin, Green FabLab, Cykelköket, Bike Kitchen North East, Material Mafia, Re-Create Design and SekundärSchick). Additionally, both Mikrofabriken and Guldminen do not organize teaching events themselves, but some of their members do. Stilbruch does not teach the general public, but they do have capacity building activities for their employees. In general, the teaching activities offered to wider audiences can be categorized into three groups (see Section 3.3.1), with some initiatives engaging in all the types.

Besides teaching, most initiatives offer public open hours, and some participate in communal events, e.g., Cykelköket has participated in many events promoting sustainable transport options, while Bike Kitchen North East members participate regularly in the critical mass rides in Berlin.

3.3.1. Teaching Activities

All FabLabs offer workshops to learn how to use the machines they have. The machine workshops are usually offered on regular intervals depending on the demand for the specific machines. For example, FabLab Berlin used to offer 3D printing workshop every week, while the CNC milling introduction was done only once a month when available. HappyLab Berlin does a general introduction to their machines once a week, free of additional charge. They offer this type of workshop to ensure the safe handling of the machinery.

In predefined workshops a certain product or task is accomplished, where the steps to get to the result and the required materials are prepared in advance. Cykelköket has done workshops with predefined tasks, e.g., how to fix a flat tire, adjust break systems or do a quick diagnose of what might need to be fixed in a used bike. These specific workshops were normally organized to train a larger volunteer group for specific events, and were not repeated regularly. FabLabs normally offer workshops on the software needed to generate models for operating their machines. Workshops are normally offered as open for anybody without previous knowledge on the subject, but highly specific courses usually attract highly specialized participants. The Green FabLab again is a bit different, they offer similar workshops, but they do not take place regularly, rather project based, and on occasions in collaboration with big companies. They also offer workshops specific to their research themes, e.g., sustainable farming and bee-hive making. Re-Create Design, SekundärSchick and Material Mafia teach different up-cycling/design workshops. Re-Create Design and Material Mafia do it on irregular intervals or per demand, whereas SekundärSchick offers several workshops per week. These events vary between sessions for school or university classes, to corporate team building activities or even consulting on specific topics, and most often take the form of predefined workshop sessions.

Bike kitchens and FabLabs engage in informal teaching sessions during their open hours. Visitors can get assistance to do their own projects, learning in unstructured sessions, either on their own or with the assistance of volunteers or other engaged visitors. In some cases, FabLabs have additional events, depending mostly on the interest of community members to offer or organize structured courses or more informal thematic meet ups.

The number of participants per workshop varies largely, even within the same initiative, starting from a pair to several dozen participants. The largest workshops were reported to have about 70 participants at the Green FabLab. However, most workshops are kept to smaller groups with an average of between five and ten people. These numbers are roughly consistent throughout the different types of workshops offered. The prices for each workshop also vary greatly even within each initiative, ranging from totally free of charge to 200–300 euros per person, for highly specialized courses at FabLabs.

*3.4. Business Aspects*

Most of the initiatives that were studied are non for profit or project-based organizations, with the exceptions being Genbyg, Re-Create Design and Microfabriken. It is not surprising that initiatives that wish to facilitate reuse and remanufacture do so with an altruistic motivation. However, the study includes three examples where these motivations are also compatible with successful commercial activities, including the longest running initiative in the study, Genbyg.

3.4.1. Financial Models

Of the fifteen initiatives analyzed, six are not dependent on external funding for their operations (although they may accept material and/or financial donations):

- Material Mafia, financed by material sales, workshops and products done on request
- Happylabs, financed by membership fees
- Mikrofabriken, financed by membership fees
- Genbyg, financed by selling secondary material and second-hand products
- Re-Create design, a private company financed by interior design projects
- SekundärSchick, financed by organizing workshops on how to remanufacture clothes

Among the other nine initiatives, four are run or supported by municipalities: Guldminen, Fixoteket, Stilbruch and Återbruket (of which the latter aims to let material sales cover all running costs i.e., rent and salaries). FabLab Berlin is financed by membership fees together with industrial sponsorship and research project funding, like the Green FabLab in Barcelona, which is partly financed by the Institute of advanced architecture of Catalonia and partly by the EU Programme Horizon 2020. Cykelköket is mainly funded by membership fees and financial support from the Swedish non-profit consortium "Studiefrämjandet" that covers their rent. Bike Kitchen North East is financed entirely by donations while the BSR exchange and donation Market is a part of the Berlin Waste Management company BSR. It is however important to notice (see Table 1) that these initiatives vary greatly with regards to what they need to finance, e.g., there are large variations in rent, staffing (if they have volunteers or not), opening hours, etc.

3.4.2. Pricing for Secondary Materials

Just five out of the fifteen analyzed projects sold secondary materials and/or products (i.e., Återbruk, Genbyg, Material Mafia, Re-Create Design, and Stilbruch). Additionally, both bike kitchens and Happy Labs Berlin sell materials and consumables at wholesale prices, not generating any income on these sales. The Green FabLab creates bee hive kits that they sell, even though they have all cutting instructions available on-line for free. In this case, Green FabLab sells the kits because people have requested it, but it is not done to generate profit.

The price of the materials for those who did sell them were calculated differently depending upon the structure of each initiative. Re-create Design takes the number of hours of work put into the material as a reference. Återbruk, Material Mafia and Stilbruch relate it to the market price of new products and use a rule of thumb to reduce the price by 50–70%. Hence the price calculation is quite incongruous but an approach that seems to work for most initiatives is the consideration and use of the market price as a benchmark.

3.4.3. Facilitating Activities v/s Providing Services

FabLabs and Bike Kitchens cater in principle to the do-it-yourself public. The personnel or volunteers that manage the open hours are there only to assist the incoming public to use the tools and advise them on their projects. However, it is often that these spaces get asked to provide a prototyping or fixing service. Depending to the policy of each initiative, they either refuse to engage (i.e., Cykelköket will not fix bikes for people so they do not compete with local bike shops) or agree to do the required service for an agreed price (i.e., Bike Kitchen North East may do the small fixed needed to get a bike functional, in order to sell the bike off for a suggested donation).

HappyLab Berlin and FabLab Berlin support prototyping but do not directly provide this service. The biggest advantage of FabLabs is their community as almost always someone associated with a FabLab has some experience in how to implement parts of the project in question. When a prototyping service is required, they may ask the people from their community is somebody wants to take up the task.

## 4. Discussion

The previous chapter presents the results of this initiative analysis, categorized in four sections: i.e., facilitating the recovery of secondary materials, facilities and tools, events and activities, and business aspects. Each of these sections have underlying themes that further describe how these initiatives operate. All these categories are collected in Table 3, to provide an overview of the results.

**Table 3.** Categorizations of the underlying themes identified among the studied initiatives.

| Section | Underlying Themes | Theme Categories |
|---|---|---|
| Facilitating the Recovery of Secondary Materials | Different recovery routes | Direct reuse |
| | | Remanufacture for others |
| | | Own product development |
| | Different material types | Wide range of materials |
| | | Miscellaneous household items |
| | | Bikes |
| | | Clothes |
| Facilities and Tools | Workshop availability to public | Closed workshops |
| | | Partially open workshops |
| | | Open workshops |
| | | None |
| | Type of tools available | Common household tools |
| | | Digital fabrication |
| | | Woodworking tools |
| | | Painting equipment |
| | | Sewing machines and other textile tools |

**Table 3.** *Cont.*

| Section | Underlying Themes | Theme Categories |
|---|---|---|
| Events and Activities | Teaching activities | Machine workshops |
| | | Predefined workshops |
| | | Informal teaching sessions |
| | Open hours and other events | |
| Business Aspects | Self-financed | Sales of materials or products |
| | | Workshop fees |
| | | Membership fees |
| | | consulting services, |
| | Externally supported | Municipal support |
| | | Industrial sponsorship |
| | | Research funding |
| | | Donations |
| | Non-profit/Commercial business | |
| | Facilitating activities v/s providing services | |

Based on the results and tacit knowledge collected in this study, the following framework is suggested to serve as a guide for establishing future CUREs: These centers should combine a secondary material warehouse with open (or partially open) workshop spaces that allow for testing and prototyping. These centers would be constitute both a physical place and a network of actors that provide experiments, and eventually reuse or remanufacture secondary materials. These centers would be expected to host capacity building activities for both the general public and professional actors, as well as support local circular economic entrepreneurship. CUREs are expected to be able to finance their operations by product and/or material sales, capacity building activities, workshop space rental, consulting services and potentially other events. Even if the design of a CURE would need to be highly dependent on the local context, more standardization of their design could be beneficial. Aspects such as keeping an inventories of available material and making material more widely available, assessing environmental impact, and exchanging experiences and learning from one another, could all improve with increased standardization and collaboration. This framework is represented in Figure 1.

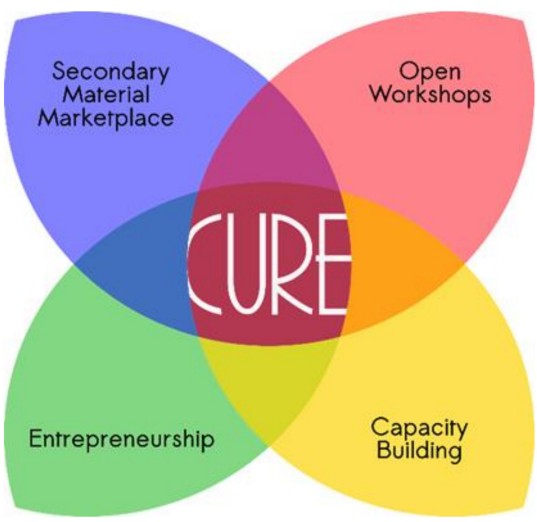

**Figure 1.** Underlying aspects that should guide future CUREs.

As stated earlier, the first two sections of the results reflect the main criteria used to select the initiatives for this study, so it is not surprising that one section is dedicated exclusively to secondary materials and another one for tools. Better collaboration and matchmaking between secondary material providers and re-manufacturers is needed in order to identify new opportunities for making something useful with the waste. Therefore, a CURE would need both a secondary material marketplace and open (or at least partially open) workshops for experimenting and increasing the matches between unused material and the people who can make use of it. Events and activities are very general terms to describe what initiatives do, but when analyzing these activities in more detail, it becomes clear that the initiatives mainly engage in teaching. This teaching is normally of a short duration and targets developing a specific skill or activity; therefore, they are more accurately described as capacity building activities, rather than general education. This capacity building is a crucial element as knowledge sharing, teaching and skills development is needed in order to both spread best practices among remanufacturers but also build awareness in the general public/consumer base. Regarding how such activities are financed, the studied initiatives are mostly externally financed, with six initiatives being economically self-sufficient. This indicates that there is a great need for development and innovation in the business models of this emerging sector in order for it to successfully scale and spread.

With all these considerations in mind, we can rephrase the main sections that describe the results of this study to better reflect what a CURE would need to incorporate. Here we describe the new main sections, and shortly discuss some aspects that seem to be relevant to consider for establishing future CURE work.

### 4.1. Secondary Material Marketplace

A warehouse where people can see and touch available secondary material is a good way to visualize how many resources can be locally recovered. Section 3.1.3. describes briefly the inconsistently between the few inventory systems described by the initiatives. This has been identified during the interviews as a mayor challenge. Currently each initiative uses material and product categories that fit their own operations, but this means that the data generated by each initiative is not comparable to any other operation, nor local waste statistics. Additionally, it is not always possible to determine the environmental impact (or benefit) that these initiatives generate, because the inventories do not necessarily account for the weight of recovered items. Therefore, it would be desirable to have a standardized system, centralized in some way, to help initiatives easily assess their impact and provide an overview of their operations, while being able to compare their material flows with other standardized statistics. Some work in this line has been done by OVAM in the Flemish region of Belgium.

### 4.2. Open Workshops

Some of the workshop spaces among the analyzed initiatives are customized for the area that they focus on, e.g., the ones working with clothing and textiles have a focus on sewing machines. Thus, an ambitious remanufacturing center that is not specialized would need a well-equipped workshop space to cover all needs. The advantage that a CURE would have by not specializing on a specific material, is that, like in a warehouse, people could get all the different materials that they need for their project in one place. Some basic equipment needed, based on the reviewed initiatives would be: common household tools (e.g., screwdrivers, pliers, cutters, hammers), wood and metal working equipment (e.g., electrically powered saws, sanders, drills) and textile tools (e.g., sewing machines). Eventually a space for finishing processes such as painting, or other surface treatments would also be desirable, although this is not crucial at the first stage.

### 4.3. Capacity Building

As described by many of the reviewed initiatives, there is no lack of public interest and the enthusiasm and driving spirit is high among the people who work for the initiatives. How all this is

optimally orchestrated is, however, still relatively unknown, and most initiatives seem to learn as they go. CUREs need to incorporate capacity building, both for individuals working at CUREs, their public, and for the CUREs themselves. The know-how that is being developed among the individuals working at the initiatives needs to be disseminated among others in the same trade. While this can (and should) happen spontaneously between people, it can also be organized, e.g., through courses, workshops, conventions, etc. Bike kitchens and FabLabs already have these type of collaborative platforms [29,30]. Another example of this is the organization Rreuse, that represents European social enterprises active in the field of reuse, repair and recycling [31]. Following these examples, it can be suggested that how CUREs cooperate and learn from each other can also be supported. Let's say that a CURE has spent a lot of time to come up with an inventory system that works well. It could be of great interest for other CUREs to know about this. They could perhaps implement the same system to make their operations more effective, and possibly also contribute to a more standardized categorization of secondary materials. The lack of orchestration of urban initiatives like the ones in this study is present also in European industrial (planned) remanufacturing. According to a market study by the European Remanufacturing Network [32], a European-level solution to encourage remanufacturing is needed to raise competitiveness. Still, urban reuse and remanufacturing is far behind industrial remanufacturing when it comes to central organization and promotion.

Because of this need for central organization and knowledge collection, CUREs would benefit from not only being a physical place but also a network of actors who can interact online. Many of the reviewed initiatives get the secondary material they need from actors in their network, and that material does not necessarily have to take a detour via a recycling center. Some initiatives provide an updated online catalog, or offer products through social media, which helps them increase their sales. Additionally, some initiatives know of large volumes of secondary material that are regularly discarded, they could potentially offer these materials available on demand through a web-page, collecting it only when an order is requested.

### 4.4. Entrepreneurship

Many of the initiatives are dependent on external support and funding, and some of them would probably need to close down without it. Overall the driving force for many of the initiatives is apparently environmental sustainability, and not 'doing business', which typically is a strong driver for industrial remanufacturing industries (e.g., [11,17,33]). Still, business needs to be done, and all the matches made between secondary materials and re-manufacturers would need to rely on some kind of enterprise to take off. Therefore, entrepreneurial support, such as that available from business incubators, would be desirable in a remanufacturing center. Additionally, it can be argued that small-scale operations focusing on improvised remanufacturing are important for increasing reuse. While planned remanufacturing and reuse typically is of a larger scale, it is predominantly carried out by larger companies which are rooted in linear operations and business logics. This makes planned remanufacture and reuse challenging to them (see [34] for an overview). Small-scale operations and start-ups are not as restricted by their heritage in that way.

### 5. Conclusions and Future Research

The aim of the present study was to report how the selected initiatives operate, how they are financed, to what extent they contribute to resource recovery and what common lessons and themes can be identified. The common themes identified have been presented in Section 2 and discussed further in Section 3, and they describe the initiatives' operations and financial models. How these initiatives contribute to resource recovery can be described qualitatively (i.e., they recovery used materials or products in a specific way), but providing an objective measure of their environmental contribution is challenging. There is a lack of an inventory system that can be easily used by these initiatives in daily operations, but that also complies with environmental analysis standards. In order to argue for the positive impact these initiatives have, it is crucial to be able to measure

their environmental performance. Two of the analyzed initiatives commented during the interviews that they had participated once in an environmental evaluation (i.e., Återbruket and Material Mafia); both evaluations were carried out by external research organizations and implied additional efforts to weigh or measure the volume of materials recovered. Återbruket operations were followed in a research project performed in 2011, showing that Återbruket helped reuse 72% (i.e., 358 tons) of the material coming in to the recycling center, while 17% of the material was sent to recycling. In comparison, a traditional recycling center during the same period in Sweden, sent on average 11% to reuse and 27% to recycling. These recirculated materials were calculated to represent a reduction of $CO_2$ equivalent emissions in 1300 tons and saved 5100MW/h of primary energy use [9]. The Material Mafia was calculated to contribute with reducing 345 tons of $CO_2$ equivalent emissions during 2015 [35].

As this study was limited to a select number of initiatives in only three urban centers it is suggested that other initiatives in more locations be identified and analyzed. Also, the lack of comparable inventory systems of all included initiatives limited the ability to quantitatively analyze environmental impact and economic viability. This needs to be considered and addressed in future research in order to more effectively inform, analyze and validate the proposed CURE model. In addition, the topics analyzed is in this paper do not nearly cover all areas that are of importance to the increase of reuse and remanufacturing. For instance, products that have been used by someone else are still rejected by many consumers, since they view such products as unreliable, dirty, and in some cases dangerous among other things [12]. Some of these worries might be handled by a well-functioning CURE, but it may also require products that are originally designed for the purpose of being reused [36]. Reuse and remanufacturing as a whole could also be supported by policies and regulations, such as the taxation of virgin materials. Nevertheless, urban reuse and remanufacturing initiatives play an important role in promoting circular consumption in society and addressing the world's waste challenge.

**Author Contributions:** Conceptualization, I.O., O.R. and S.H.; Data curation, I.O., S.H. and L.U.; Formal analysis, I.O., O.R., S.H. and L.U.; Funding acquisition, I.O.; Investigation, I.O., S.H. and L.U.; Methodology, I.O. and O.R.; Validation, I.O. and O.R.; Writing—original draft, I.O., S.H. and L.U.; Writing—review & editing, I.O., O.R. and S.H.

**Funding:** This research was funded by EIT Climate-KIC, task ID TC2018B_4.4.3-CURE_P183-1A.

**Acknowledgments:** The authors wish to thank all the initiatives that so generously contributed with their time to answer our questions and shared their knowledge with us for this investigation. Without their collaboration this research would not have been possible.

**Conflicts of Interest:** The authors declare no conflict of interest.

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
