# Peer review of "Designing Away Waste: A Comparative Analysis of Urban Reuse and Remanufacture Initiatives"

_recycling, doi:10.3390/recycling4020015_

Round 1

Reviewer 1 Report

According to the UN, about 720 billion tons of production and consumption wastes are generated each year, of which 440 billion are in developed countries. In the world there is a waste recycling industry, landfills based on manual labor. In countries with a high level of environmental awareness, the main indicator of civilized attitudes towards garbage is the proportion of waste that is recycling. The most important criterion for achieving this goal is to incinerate a minimum, recycle a maximum.

Countries with a high standard of living are dumping landfills in third world countries, where they pay less attention to safety and environmental problems. Countries with a high ecological culture of citizens and politicians can spend money on high-tech equipment, while poor countries with a low standard of living suffer from their waste and from those exported from other countries both in a legal and illegal manner. But in the end, it becomes worse for all the inhabitants of the Earth: air and water are common to all.

 We need a unified global policy on waste management, laws at the level of the G20 countries, technologies for recycling. Ideally, international programs are needed that fund waste recycling and monitor the situation in the waste recycling industry and landfills.

The article does not reflect the globalization of problems of production and consumption wastes. The level of scientific achievements for solving the problem of production and consumption wastes is not shown in the paper.

In our opinion, the authors should either expand the introduction, or add a separate section of the article showing the global nature of the problems and the level of development of science to solve emerging problems.

Author Response

p { margin-bottom: 0.1in; line-height: 120%; }

Thank you for your comments. The authors agree that bringing in the global dimension of the waste and consumption problem is a highly relevant aspect for this topic and hope to have been able to address it in a understandable manner. The introduction is now structured with several sub-sections, one of which refers to the global nature of the problem, linking the global aspect of current trade and production to local solutions to waste management.

Two major changes have been made to the manuscript: (1): More references have been added, both in the introduction and the discussion, to clarify the study’s relation to previous research, and (2): The manuscript was written in a way that could lead readers to expect a full literature review. This was not our intention, so now it is more clearly stated that it is a comparative analysis of urban remanufacture and reuse initiatives. This is also reflected in the manuscripts new title and abstract.

Reviewer 2 Report

The article addresses an interesting topic, from an original approach. However, it has some deficiencies that should be improved.

Thus, in a work of these characteristics, in which a review on a subject is made, based on the analysis of several real cases, the literature review should be considerably broader (there are barely 20 references for an article of these characteristics).

There are complete paragraphs in which there are hardly any references, even in the introduction section, where the research is to be contextualized. For example (only as an example, since there are many more cases that should be corrected), the introduction mentions terms such as "planned manufacturing" or "improvised manufacturing", without mentioning a bibliographic reference. And this is worsened by the absence of a review of literature section, so the introduction should conduct a much deeper analysis.

The work is structured around 4 axes or main themes, which should also be widely analyzed in the literature review section. In addition, it would be convenient to adequately justify why these four criteria are selected, with the reinforcement, once again, of the literature. This is only done, very slightly, in the discussion section, but it is not enough.

The methodology section appears almost at the end of the work. It would be more advisable that it be right after the literature review, because it would help to contextualize the work done better.

Although some explanations are given, a broader justification (beyond those already mentioned in the text) of the reasons that led to the selection of the 15 cases of analysis would be necessary.

No limitations on work are mentioned, which would be helpful to propose future lines of research. One of the limitations could be the geographic context excessively centered in certain places, which could make it difficult to extrapolate the results to other different environments. In addition, the criteria that are argued for the selection of cases should be reinforced with literature.

In line 417 a work is mentioned that does not appear in the list of references, in the final section. Please adapt the work to the standards of the editorial.

Some presentation errors must be corrected in Table 2, as well as the name in Figures 1 and 2 (in the latter, it appears repeated).

Author Response

Thank you for your comments.

A point-by-point response is available in the attached document.

Reviewer 3 Report

This manuscript addresses the important challenge of promoting product reuse and remanufacturing in order to advance a circular economy and reduce material throughput. It is well written with no grammar or language issues. Its main strengths are: a) the data collected from 15 remanufacturing centers, and b) the discussion and analysis of results in section 3 on pages 11 and 12. The paper, however, needs a major revision before it can be published due to the following weaknesses:

1)      The introduction must more clearly state the research goals (besides just looking for themes) and the methods employed.

2)      The literature review currently is very brief and inadequate; I suggest expanding to include some of the market barriers to remanufacturing, the role of entrepreneurs, the need for policy action (e.g., to tax resources instead of labor), etc. I’ve included some suggested articles below.

3)      After the literature review the authors should include the Methods section (which is currently included on page 12 as section 4). This section must also include additional information such as how many interviews where completed, who the interviewees were, length of each interview and how was it conducted (in person or by phone), how were they coded and analyzed, etc.

4)      The Results section should begin with some discussion of the results and not Table 1. I also recommend moving Table 1 to an appendix.

5)      I like the four areas presented in Figure 1 and would recommend organizing a table with these themes as it will be more helpful for the reader.

6)      The discussion and conclusions sections are the strongest; I suggest also including a brief discussion about the limitations of the study and linking back to key findings from the literature review (when developed).

Overall this is a good paper but it has too much technical detail which is not clearly organized and makes it challenging for the reader to follow through. It needs a major revision to better state the goals and the methods, better organize, communicate and link the key findings to previous research and potential policy and business actions.

Some helpful publications to consider when revising your paper include:

·         Abbey J, M. Meloy, J. Blackburn and V. Daniel Guide Jr., “Consumer markets for remanufactured and refurbished products”, California Management Review, Vo. 57, No.4, Summer 2015, pp. 26-42.

·         European Commission, 2015, “Remanufacturing Market Study”, A report by the partners of ERN, funded by Horizon 2020, 2015, https://www.remanufacturing.eu/wp-content/uploads/2016/01/study.pdf

·         Guide D. & L. Jiayi, 2010, “The potential for cannibalization of new products sales by remanufactured products”, Decision Sciences, Vol. 41, Issue 3, pp. 547-572.

·         Matsumoto, M. 2009. “Business frameworks for sustainable society: a case study on reuse industries in Japan”, Journal of Cleaner Production, 17, 1547-1555.

·         Matsumoto M. and Y. Umeda, 2011, “An analysis of remanufacturing practices in Japan”, Journal of Remanufacturing, 1:2, http://journalofremanufacturing.springeropen.com/articles/10.1186/2210-4690-1-2

·         Pearce II, J. 2009. “The Profit-making Allure of Product Reconstruction,” MIT Sloan Management Review 50.3, Spring 2009, pp. 59-65.

·         Subramoniam, R., D. Huisingh, and R. Chinnam, 2010, “Aftermarket remanufacturing strategic planning decision-making framework: theory & practice”, Journal of Cleaner Production, 18: 1575-1586.

·         Veleva V., and G. Bodkin, 2017, “Emerging drivers and business models for equipment reuse and remanufacturing in the U.S.: Lessons from the biotech industry”, Journal of Environmental Planning & Management, September, http://www.tandfonline.com/doi/full/10.1080/09640568.2017.1369940

·         Abbey J, M. Meloy, J. Blackburn and V. Daniel Guide Jr., “Consumer markets for remanufactured and refurbished products”, California Management Review, Vo. 57, No.4, Summer 2015, pp. 26-42.

Author Response

p { margin-bottom: 0.1in; direction: ltr; color: rgb(0, 0, 0); line-height: 120%; }p.western { font-family: "Liberation Serif", "Times New Roman", serif; font-size: 12pt; }p.cjk { font-family: "Noto Sans CJK SC Regular"; font-size: 12pt; }p.ctl { font-family: "FreeSans"; font-size: 12pt; }

Thank you very much for your very relevant literature suggestions and overall comments. The authors feel that it has improved the article immensly and are very happy with the results.

Please find a point-to-point response below.

This manuscript addresses the important challenge of promoting product reuse and remanufacturing in order to advance a circular economy and reduce material throughput. It is well written with no grammar or language issues. Its main strengths are: a) the data collected from 15 remanufacturing centers, and b) the discussion and analysis of results in section 3 on pages 11 and 12. The paper, however, needs a major revision before it can be published due to the following weaknesses:

1)     The introduction must more clearly state the research goals (besides just looking for themes) and the methods employed.

The research goals is more specified in the end of the introduction, and is now followed by the method section, which hopefully will make this clearer.

2)     The literature review currently is very brief and inadequate; I suggest expanding to include some of the market barriers to remanufacturing, the role of entrepreneurs, the need for policy action (e.g., to tax resources instead of labor), etc. I’ve included some suggested articles below.

The manuscript was written in a way that could lead readers to expect a full literature review. This was not our intention, so now it is more clearly stated that it is a comparative analysis of urban remanufacture and reuse initiatives. This is also reflected in the manuscripts new title and abstract.

However, the articles suggested by the reviewer were very interesting and highly relevant and have to a large extent been included, both in the introduction and the discussion, to clarify the study’s relation to previous research

3)     After the literature review the authors should include the Methods section (which is currently included on page 12 as section 4). This section must also include additional information such as how many interviews where completed, who the interviewees were, length of each interview and how was it conducted (in person or by phone), how were they coded and analyzed, etc.

The method section has been moved, it now follows the introduction section. The authors also considered this to be a more adequate placing, but the template of the journal had the methodology at the end of the work and this was followed in the first submission. The authors expect that the editor makes a final decision on this matter.

Regardless, the methods section has been expanded to include the additional information requested.

4)     The Results section should begin with some discussion of the results and not Table 1. I also recommend moving Table 1 to an appendix.

Thank you for the suggestion. We considered doing this but felt that the overview table 1 provides is needed in order to better understand the remainder of the results section. We have however placed initial discussion before the table.

5)     I like the four areas presented in Figure 1 and would recommend organizing a table with these themes as it will be more helpful for the reader.

Figure 1 has been changed into a table, for the ease of the reader.

6)     The discussion and conclusions sections are the strongest; I suggest also including a brief discussion about the limitations of the study and linking back to key findings from the literature review (when developed).

Study limitations have been added in the introduction and later again briefly discussed in the conclusion section.

Overall this is a good paper but it has too much technical detail which is not clearly organized and makes it challenging for the reader to follow through. It needs a major revision to better state the goals and the methods, better organize, communicate and link the key findings to previous research and potential policy and business actions.

Some helpful publications to consider when revising your paper include:

·        Abbey J, M. Meloy, J. Blackburn and V. Daniel Guide Jr., “Consumer markets for remanufactured and refurbished products”,California Management Review, Vo. 57, No.4, Summer 2015, pp. 26-42.

·        European Commission, 2015, “Remanufacturing Market Study”, A report by the partners of ERN, funded by Horizon 2020, 2015, https://www.remanufacturing.eu/wp-content/uploads/2016/01/study.pdf

·        Guide D. & L. Jiayi, 2010, “The potential for cannibalization of new products sales by remanufactured products”,Decision Sciences, Vol. 41, Issue 3, pp. 547-572.

·        Matsumoto, M. 2009. “Business frameworks for sustainable society: a case study on reuse industries in Japan”, Journal of Cleaner Production, 17, 1547-1555.

·        Matsumoto M. and Y. Umeda, 2011, “An analysis of remanufacturing practices in Japan”, Journal of Remanufacturing, 1:2, http://journalofremanufacturing.springeropen.com/articles/10.1186/2210-4690-1-2

·        Pearce II, J. 2009. “The Profit-making Allure of Product Reconstruction,” MIT Sloan Management Review 50.3, Spring 2009, pp. 59-65.

·        Subramoniam, R., D. Huisingh, and R. Chinnam, 2010, “Aftermarket remanufacturing strategic planning decision-making framework: theory & practice”, Journal of Cleaner Production, 18: 1575-1586.

·        Veleva V., and G. Bodkin, 2017, “Emerging drivers and business models for equipment reuse and remanufacturing in the U.S.: Lessons from the biotech industry”,Journal of Environmental Planning & Management, September,http://www.tandfonline.com/doi/full/10.1080/09640568.2017.1369940

·        Abbey J, M. Meloy, J. Blackburn and V. Daniel Guide Jr., “Consumer markets for remanufactured and refurbished products”,California Management Review, Vo. 57, No.4, Summer 2015, pp. 26-42.

Round 2

Reviewer 1 Report

I wish the authors success in solving the problem of global distributed production of goods in this world and local reusing,recycling,recovering materials and reuse of waste/

Author Response

The authors than the reviewer for their support and valuable feedback.

Best regards,

Isabel

Reviewer 2 Report

Article has greatly improved over the previous version. In this new version new references have been introduced, the variables are better explained, the methodology is better understood and makes more sense, and the limitations of the work are detailed.

I am satisfied with the changes made by the authors.

Only one important aspect. The version that the authors have uploaded includes comments and modifications, which makes it very difficult to read the paper.

Author Response

Thank you very much for this positive feedback.

We have uploaded the version with the modifications, because the editor´s instructions was to use track changes on the document so that the modifications on the text were easily identified.

We will however submit also a "clean" version without comments and with all the changes accepted in the next submission.

Reviewer 3 Report

Clearly improved manuscript with now well-developed Methods section, improved results and discussion sections and addition to some more references and limitations of the study. There are, however, several areas that should be improved.

First, the authors should more clearly state the goals of the study right in the introduction (e.g., they can use the first paragraph in Section 5 (“The aim of the present study was to report how the selected initiatives operate, how they are financed, to what extent they contribute to resource recovery and what common lessons and themes can be identified…”)

Second, the authors must make it much clearer why it’s important to advance reuse and remanufacturing. For instance, they can include data that shows the positive impacts on climate change from reuse and remanufacturing compared to recycling or landfill/waste to energy. This should be included in the introduction. I also suggest to elaborate a bit more on the recent EU directive to advance reuse and remanufacturing/repair activities.

Third, while I understand the authors do not aim to provide a comprehensive literature review of reuse and remanufacturing, I still feel strong about including at least a page about this, with summary of the key drivers and barriers to remanufacturing/reuse identified in the literature.

Fourth, Figure 4 should not be in the conclusion and certainly not the last thing in the article! I suggest moving to the Discussion section. The latter should also include some policy suggestions (e.g., taxing raw materials, providing more time and other incentives for people to use repair cafes, etc.)

Overall much improved manuscript which can easily be strengthened further.

Author Response

Thank you again for your relevant comments.

Please find attached a poin-to-point response to your suggestions.

We are very greatful for your feedback and consider that the article has improved substantially.

Best regards,

Isabel
